

# Development and validation of a novel survival model for acute myeloid leukemia based on autophagy-related genes

Li Huang, Lier Lin, Xiangjun Fu and Can Meng

Department of Hematology, Hainan General Hospital (Hainan Affiliated Hospital of Hainan Medical University), Haikou, China

## ABSTRACT

**Background.** Acute myeloid leukemia (AML) is one of the most common blood cancers, and is characterized by impaired hematopoietic function and bone marrow (BM) failure. Under normal circumstances, autophagy may suppress tumorigenesis, however under the stressful conditions of late stage tumor growth autophagy actually protects tumor cells, so inhibiting autophagy in these cases also inhibits tumor growth and promotes tumor cell death.

**Methods.** AML gene expression profile data and corresponding clinical data were obtained from the Cancer Genome Atlas (TCGA) and Gene Expression Omnibus (GEO) databases, from which prognostic-related genes were screened to construct a risk score model through LASSO and univariate and multivariate Cox analyses. Then the model was verified in the TCGA cohort and GEO cohorts. In addition, we also analyzed the relationship between autophagy genes and immune infiltrating cells and therapeutic drugs.

**Results.** We built a model containing 10 autophagy-related genes to predict the survival of AML patients by dividing them into high- or low-risk subgroups. The high-risk subgroup was prone to a poorer prognosis in both the training TCGA-LAML cohort and the validation GSE37642 cohort. Univariate and multivariate Cox analysis revealed that the risk score of the autophagy model can be used as an independent prognostic factor. The high-risk subgroup had not only higher fractions of CD4 naïve T cell, NK cell activated, and resting mast cells but also higher expression of immune checkpoint genes *CTLA4* and *CD274*. Last, we screened drug sensitivity between high- and low-risk subgroups.

**Conclusion.** The risk score model based on 10 autophagy-related genes can serve as an effective prognostic predictor for AML patients and may guide for patient stratification for immunotherapies and drugs.

## INTRODUCTION

Acute myeloid leukemia (AML) is a kind of malignant blood cancer, accounting for about 1% of all cancers (*Molica et al., 2019*; *Winer & Stone, 2019*; *Moors et al., 2019*). AML is

Corresponding author
Lier Lin,
linlie123456202010@163.com

characterized by impaired hematopoietic function and bone marrow (BM) failure, leading to fatal consequences due to the clonal expansion of undifferentiated myeloid progenitor cells (*Cai & Levine, 2019*; *Hunter & Sallman, 2019*; *Gill, 2019*). Autophagy is an important biological process, vital to survival, differentiation, development, and homeostasis, and can play a very important role in tumors. Under normal circumstances, autophagy can inhibit the early development of cancer (*Onorati et al., 2018*; *Glick, Barth & Macleod, 2010*; *Mizushima & Komatsu, 2011*; *Li et al., 2017*) by eliminating damaged proteins and organelles and reducing cell damage and chromosome instability. However, under hypoxic or low nutritional conditions, tumors can obtain nutrients through autophagy (*Boya et al., 2016*; *Kim & Lee, 2014*; *Fan et al., 2020*; *Parzych & Klionsky, 2014*). Recent studies found that inhibiting autophagy effectively inhibits tumor growth and promotes tumor cell death (*Luan et al., 2019*; *Wang et al., 2019*; *Liang et al., 2020*). Moreover, autophagy-related gene signatures can effectively predict the clinical outcome of pancreatic ductal adenocarcinoma and breast tumors, but the research on autophagy prognostic biomarkers of AML is still insufficient.

In this study, we used AML data from the TCGA database (TCGA-LAML) and the GEO database (GSE37642). We obtained 35 prognosis-related autophagy genes in the TCGA data and used 10 of those to construct a prognostic model and then verified it through the GEO database. Our model had good predictive performance suggests that these 10 autophagy genes may be related to the tumor microenvironment and could provide new insights for the therapeutic strategies and prognosis of AML.

## MATERIALS AND METHODS

### Database
The TCGA-LAML dataset ($n = 200$) was obtained from the TGCA database (https://portal.gdc.cancer.gov/). After deleting data with imperfect clinical information, we included the remaining 140 patients in the study. The GSE37642 dataset was obtained from the GEO database (https://www.ncbi.nlm.nih.gov/geo/query/acc.cgi?acc=GSE37642), and we specifically used the two datasets GSE37642- GPL96 and GSE37642- GPL570. After merging ($n = 562$), we used "sva" R package to eliminate any batch effects (*Varma, 2020*; *Leek & Storey, 2007*; *Leek et al., 2012*). The TCGA-LAML cohorts were the training group, the GSE37642 cohorts were the verification group. The autophagy gene set (Table S1) was obtained from the autophagy database (http://www.autophagy.lu/).

### Autophagy signature construction and validation
Autophagy-related genes were extracted from TCGA-LAML, and univariate Cox analysis was used, with $p < 0.05$ considered significant. Next, we performed LASSO analysis and multivariate Cox to obtain the most critical prognostic genes, and then construct an autophagy model. The LASSO coefficients (β) as follows:

Risk Score = (βmRNA1 ×expression level of mRNA1) + (βmRNA2 ×expression level of mRNA2) + ⋯ + (βmRNAn ×expression level of mRNAn) (*Livingston et al., 2016*; *Apfel et al., 1999*; *Toulopoulou et al., 2019*).

The $\beta$ in this formula refers to the regression coefficient. The GSE37642 data set was used as a validation 1 cohort. In addition, we further verified the reliability of the prognostic gene signature by randomly dividing the training set (TCGA-LAML) into a verification 2 cohort and a verification 3 cohort. The autophagy risk score of each patient was calculated according to the uniform formula determined in the training cohort. We determine the best autophagy risk scoring standard through the "survminer" software package (*Walter, Sánchez-Cabo & Ricote, 2015*), and then divide the patients into high- and low-risk groups. In addition, we also constructed a prognostic nomogram.

### Estimation of immune cell type fractions

The CIBERSORT algorithm is used to estimate the immune cell types of TCGA data (*Alaa et al., 2019*; *Gentles et al., 2015*; *Newman et al., 2019*; *Chen et al., 2018*).

### Generation of immunescore and stromalscore

The ESTIMATE package (*Yoshihara et al., 2013*) was used to estimate the ratio of immune-stromal components in each sample in the tumor microenvironment in the form of two kinds of scores: Immune Score, and Stromal Score, which positively correlate with the ratio of immune and stroma, respectively. Meaning the higher the respective score, the larger the ratio of the corresponding component in the tumor microenvironment.

### Functional enrichment analysis

The Kyoto Encyclopedia of Genes and Genomes (KEGG) and Gene Ontology (GO) analysis of all differentially expressed genes (DEGs) by R software with $p < 0.01$ set as the threshold. Gene Set Enrichment Analysis (GSEA software, version 4.0.1) was used to investigate the pathways enriched in the high-risk subgroups. The number of random sample permutations was set at 10.

### Statistical analysis

LASSO analysis was performed using the "glmnet" package (*Engebretsen & Bohlin, 2019*; *Blanco et al., 2018*). The number of folds used in cross-validation was 10. The Time-dependent receiver operating characteristic (ROC) curve was used to evaluate the predictive performance of 10-gene features. The area under the ROC curve (AUC) was calculated by using the "survivalROC" package (*Le et al., 2020*; *Do & Le, 2020*; *Li et al., 2021*; *Le et al., 2021*). The decision curve analysis was carried out using the "rmda" software package. The "rms" software package was used for nomogram and calibration diagrams. We use one-way ANOVA to analyze multiple sets of normalized data. All statistical analyses were performed using R software (version 3.5.1) and GraphPad Software (version 7.00). $p < 0.05$ is considered statistically significant.

## RESULTS

### Establishing an autophagy-related model and functional enrichment analysis

Thirty-five autophagy genes were related to prognosis in TCGA (Fig. 1A), and LASSO regression analysis narrowed down the list (Figs. 1B, 1C), to include 10 autophagy genes

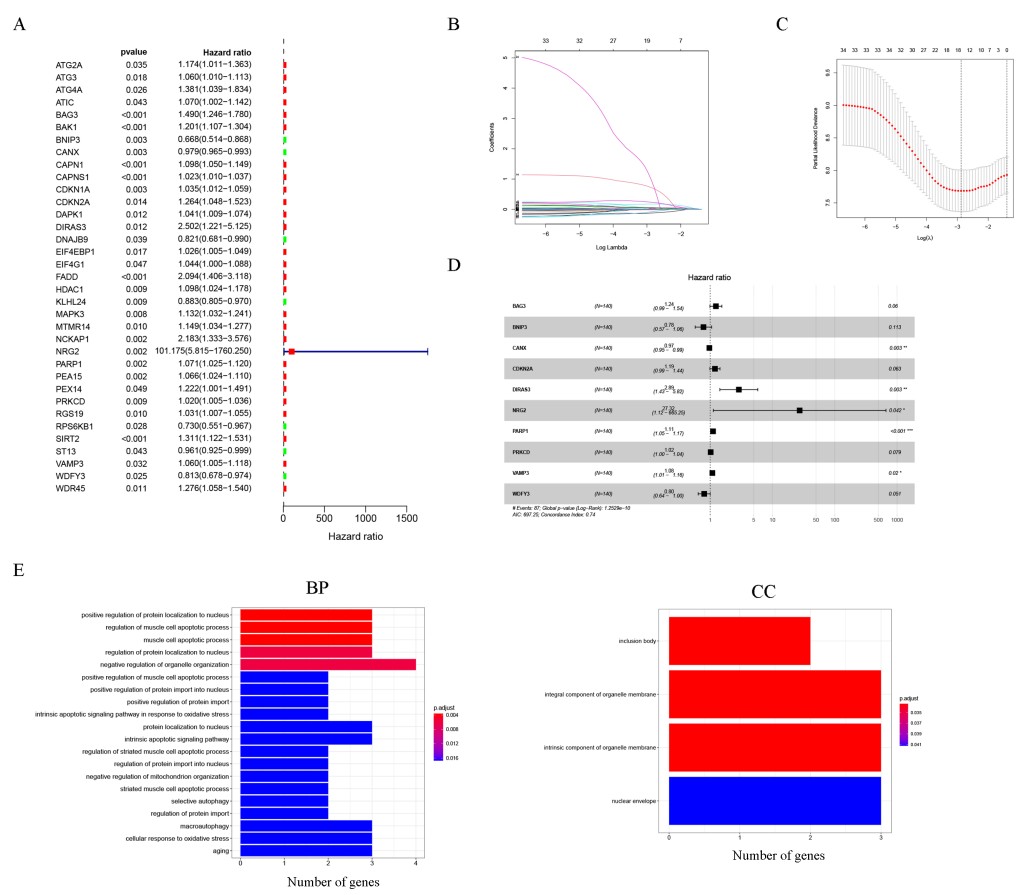

**Figure 1  Construction of the autophagy model.** (A) Univariate Cox analysis results of the TCGA-LAML cohort. (B) LASSO coefficients of autophagy-related genes. Each curve represents an autophagy gene. (C) 1,000-fold cross-validation for variable selection in the LASSO regression *via* 1-SE criteria. (D) Multivariate Cox analysis results. (E) GO analysis results.

(*BAG3*, *BNIP3*, *CANX*, *CDKN2A*, *DIRAS3*, *NRG2*, *PARP1*, *PRKCD*, *VAMP3*, *WDFY3*) for prognostic model construction (Fig. 1D).

The GO results indicated that 10 autophagy genes were significantly enriched in the biological process (BP) and cellular components (CC) categories (Fig. 1E), such as positive regulation of protein localization to nucleus, regulation of muscle cell apoptotic process, muscle cell apoptotic process, regulation of protein localization to nucleus, negative regulation of organelle organization, positive regulation of muscle cell apoptotic process, positive regulation of protein import into nucleus, positive regulation of protein import, intrinsic apoptotic signaling pathway in response to oxidative stress, protein localization to nucleus, intrinsic apoptotic signaling pathway, regulation of striated muscle cell apoptotic process, regulation of protein import into nucleus, negative regulation of mitochondrion organization, striated muscle cell apoptotic process, selective autophagy, regulation of protein import, inclusion body, integral component of organelle membrane, intrinsic

component of organelle membrane, and nuclear envelope. In addition, it is worth noting that the results of the KEGG analysis did not enrich for obvious pathways.

## Evaluation of autophagy risk score

After dividing patients into high-risk and low-risk subgroups, we found an important result that the high-risk group was significantly associated with poor prognosis in the TCGA-LAML cohort ($P = 6.975e-09$; Fig. 2A). The AUC of the one-, three-, and five-year overall survival (OS) in the TCGA-LAML cohort were 0.819, 0.846, and 0.887, respectively (Fig. 2B). Compared with the other six signatures (*Chen et al., 2020*), our signature showed a higher C-index (0.7240) and AUCs for one-, three-, and five-year OS predictions (Figs. 2C, 2D).

In order to verify the predictive value of the 10-gene signature, we calculated the risk scores of patients in the GSE37642 cohort (validation 1 set). We found that the results of the GSE37642 cohort were consistent with the results in the TCGA cohort, and the OS of the high-risk group was significantly lower than that of the low-risk group ($P < 0.001$). The AUCs for one-, three-, and five-year OS were 0.638, 0.553, and 0.532, respectively (Fig. 2F). In addition, we further verified the reliability of the model. We randomly dividing the training set into a verification 2 set (Figs. S1A–S1D) and a verification 3 set (Figs. S1E–S1H), the signature had reliable predictive ability (Fig. S1). Taking this together, the 10-gene signature was capable of predicting OS in AML. The clinical information of the patients was shown in Table S2.

## Clinical correlation analysis

Univariate and multivariate COX analysis of clinically relevant factorsshowed that age ($p < 0.001$) and riskScore ($p < 0.001$) were independent prognostic indicators in the TCGA-LAML cohort (Figs. 3A, 3B), and that age ($p < 0.001$), runx1-mutation ($p < 0.001$), and riskScore ($p = 0.019$) were independent prognostic indicators in the GSE37642 cohort (Figs. 3C, 3D).

## Nomogram analysis results of TCGA-LAML cohort and GSE37642 cohort

In order to better evaluate the relationship between genes and prognosis in the model, we used a nomogram to analyze it. The results show that in the TCGA-LAML cohort, *BNIP3*, *CANX*, and *WDFY3* have a positive correlation with OS, and B*AG3*, *CDKN2A*, *DIRAS3*, *NRG2*, *PARP1*, *PRKCD*, and *VAMP3* have a negative correlation with OS (Fig. 4A). In addition, in the GSE37642 cohort, *CANX*, *CDKN2A*, *NRG2*, and *VAMP3* have a positive correlation with OS, and *BAG3*, *BNIP3*, *DIRAS3*, *PARP1*, *PRKCD*, and *WDFY3* have a negative correlation with OS (Fig. 5A). The calibration plots showed that the nomogram could accurately predict the one-, three-, and five-year OS (Figs. 4B–4D, Figs. 5B–5D) with a harmonious consistency (TCGA-LAML, C-index = 0.72; GSE37642, C-index = 0.66) between the predicted and observed survival.

## Significant differences between high- and low-risk subgroups

The patients were scored by autophagy-related gene models, and the patients were divided into high- and low-risk groups based on the optimal score. Principal components analysis

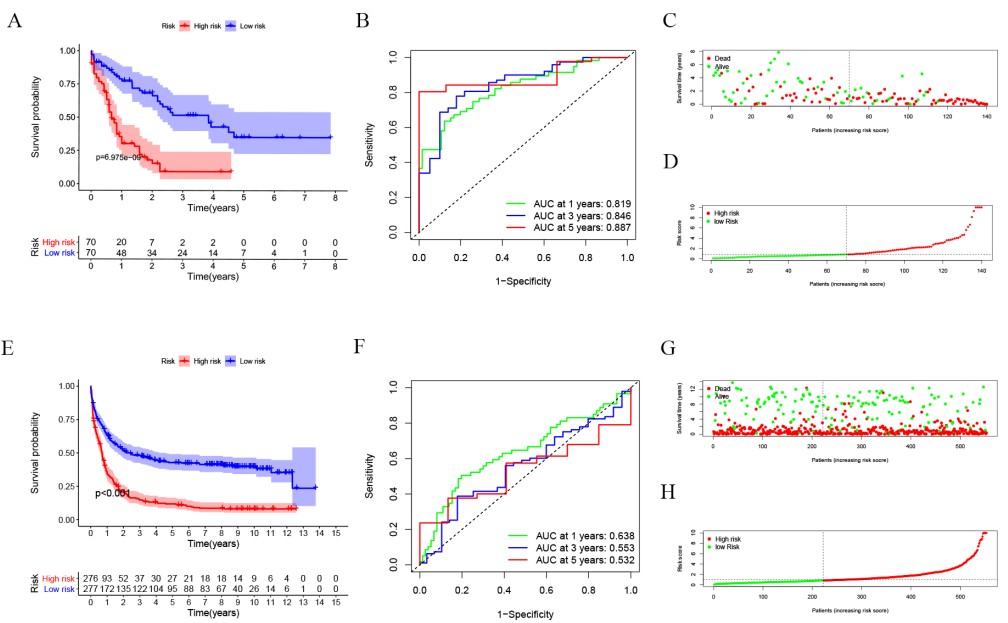

**Figure 2 Evaluation of Autophagy Risk Score.** Kaplan–Meier curve of the prognostic model in the TCGA-LAML cohort (A) and GSE37642 cohort (E). Time-dependent ROC analysis for one-, three-, and five-year overall survival (OS) of a prognostic model in the TCGA-LAML cohort (B) and GSE37642 cohort (F). The distribution of the survival status of patients in the TCGA-LAML cohort (C) and GSE37642 cohort (G). The distribution of risk score in survival outcome analysis for TCGA-LAML cohort (D) and GSE37642 cohort (H).

(PCA) supports the classification of AML patients into two subgroups (Fig. 6A). In order to further analyze the difference between the high-risk and low-risk subgroups, the ESTIMATE algorithm was used to analyze the TCGA-LAML tumor microenvironment. The results showed that high ImmuneScore was significantly associated with poor survival (Fig. 6B). Another important finding was that ImmuneScore and StromalScore were higher in the high-risk group (Fig. 6C). In addition, age was significantly correlated with both Immune Score and Stromal Score (Fig. 6D).

In order to explore the differences in immune infiltrating cells in the high- and low-risk subgroups, we used the CIBERSORT algorithm to analyze the composition of 22 immune cells in the TCGA-LAML cohort (Fig. S2) and analyzed the correlation between different immune infiltrating cells (Fig. S3). In addition, the difference in immune infiltrating cells between high and low-risk subgroups is shown in Fig. 6E. Further analysis showed that the high expression of mast cells resting was associated with a better prognosis and the NK cells activated with high expression was associated with a poor prognosis (Fig. 6F).

PDL1 (CD274) and CTLA4 play a very important role in the immunotherapy of AML. We found that the high-risk group had higher expression levels of PDL1 and CTLA4 (Fig. 6G). GSEA analysis results showed that KEGG CHEMOKINE SIGNALING PATHWAY, KEGG CELL ADHESION MOLECULES CAMS, KEGG CYTOKINE CYTOKINE RECEPTOR INTERACTION, KEGG HEMATOPOIETIC CELL LINEAGE, and KEGG INTESTINAL IMMUNE NETWORK FOR IGA PROC were enriched in the high-risk group (Fig. 6H).

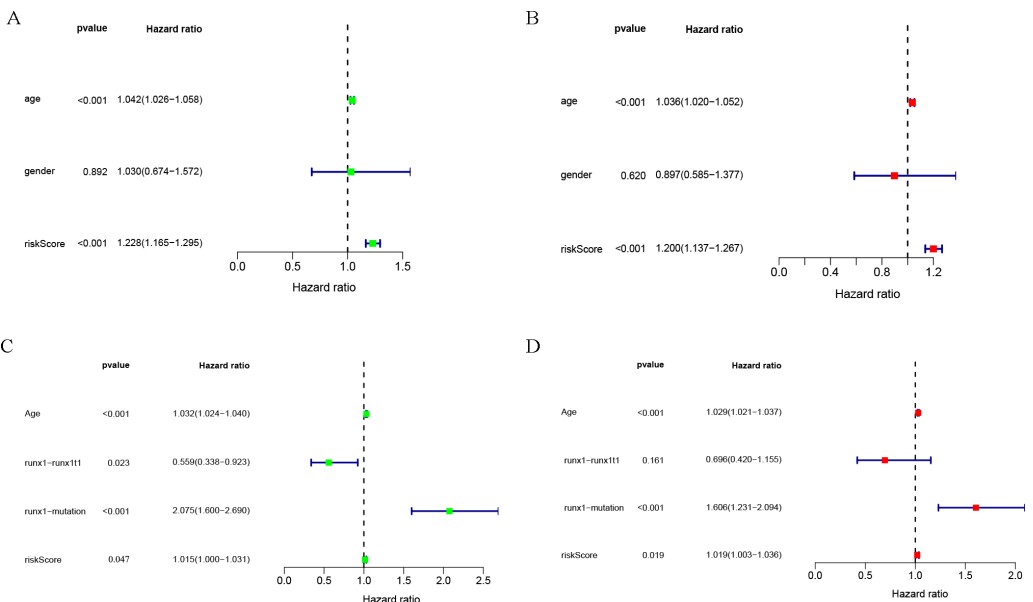

**Figure 3** **Clinical correlation analysis.** Forest plot of the univariate (left) and multivariate (right) Cox regression analysis in the TCGA-LAML cohort (A, B), and GSE37642 cohort (C, D) for acute myeloid leukemia (AML).

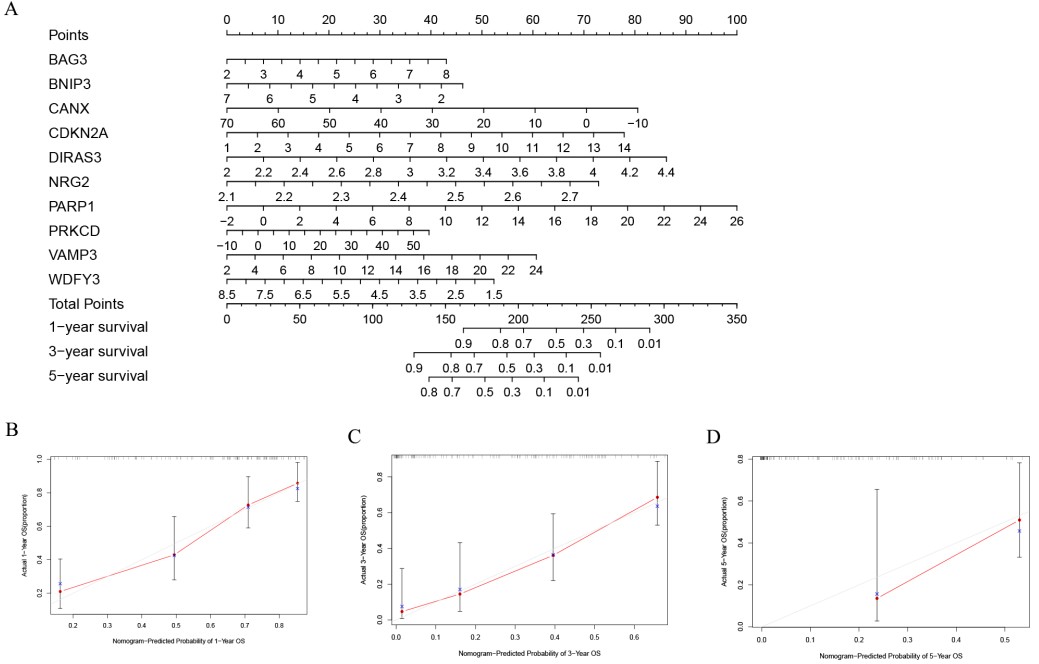

**Figure 4** **Construction of a nomogram based on the 10 hub genes.** (A) Construction of the nomogram in the TCGA cohort. (B–D) Calibration maps used to predict the 1-year (B), 3-year (C), and 5-year survival (D).

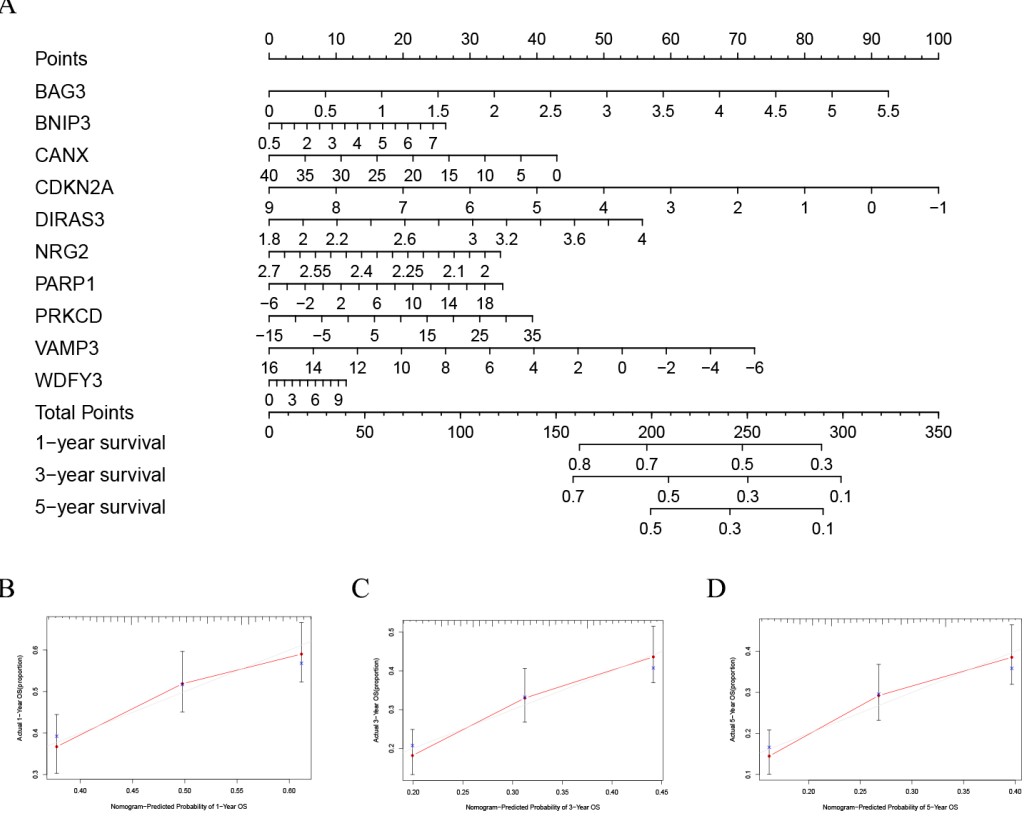

**Figure 5** **Validation of a nomogram based on the 10 hub genes.** (A) Validation of the nomogram in the GSE37642 cohort. (B–D) Calibration maps used to predict the one–year (B), three–year (C), and five–year survival (D).

The results of drug sensitivity analysis showed that there are significant differences between 24 chemotherapy drugs between high-risk and low-risk patients, which may provide help for personalized treatment of AML patients (Fig. 7).

## DISCUSSION

Autophagy has been shown to play an important role in the occurrence and development of tumors, especially in AML (*Yun & Lee, 2018*; *Fan et al., 2019*; *Levy, Towers & Thorburn, 2017*; *Zhang et al., 2019*). Targeting autophagy can overcome the chemoresistance of acute myeloid leukemia (*Piya, Andreeff & Borthakur, 2017*), granulocytic AML differentiation relies on non-canonical autophagy pathways, and restoring autophagic activity might be beneficial in differentiation therapies (*Wu et al., 2019*; *José-Enériz et al., 2019*; *Jin et al., 2018*). CXCR4-mediated signal-regulated autophagy can also affect the survival and drug resistance of acute myeloid leukemia cells (*Hu et al., 2018*).

In this study, we first identified 10 autophagy genes related to AML patients' prognosis from the training group through univariate COX analysis, LASSO regression analysis, and multivariate COX analysis, to establish a risk score model. According to the optimal value of
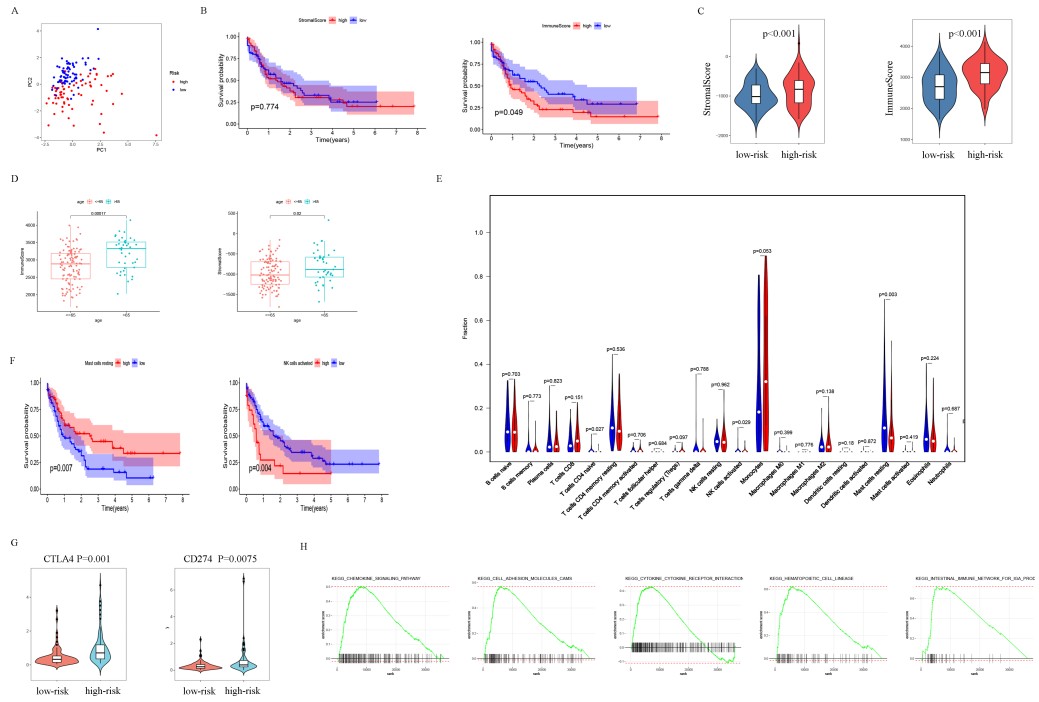

**Figure 6** **Analysis of differences between high- and low-risk subgroups (tumor microenvironment, immune cell infiltration, immune checkpoint regulators, and GSEA analysis).** (A) PCA analysis supported the stratification into two AML subclasses (high-risk (red) and low-risk (blue) groups) in TCGA cohort. (B) The survival for subgroups with different stromalscore (left) and immunescore (right). (C) The high-risk group has a higher ImmuneScore and StromalScore. (D) Age has a significant correlation with both ImmuneScore and StromalScore. (E) The comparison of immune cell fractions between high- and low-risk subgroups. (F) A high-level of mast cells resting is significantly associated with better survival, a high-level NK cells activated is significantly associated with poor survival. (G) CTLA4 and CD274 have higher expression levels in the high-risk group. (H) The pathways enriched in the high-risk group through GSEA analysis.

risk score, patients were divided into high- and low-risk subgroups. In the training group, a high-risk score was significantly correlated with poor prognosis ($p = 6.975e-09$). Then we conducted verification in the GSE37642 cohort, and the results supported that high-risk subgroups were significantly more related to poor prognosis ($p < 0.001$). Next, we tested the accuracy of the model, and the results showed that the predictive performance of the model was good (Figs. 2B, 2F). Interestingly, there was a tendency of shorter survival in patients with higher risks in TCGA data but not in GSE37642 (Figs. 2C, 2D, 2G, 2H). Testing with clinically relevant factors indicates that risk score in our model is an independent factor for AML in both TCGA-LAML and GSE37642 cohorts. Furthermore, the nomogram displayed the correlation between one-, three-, and five-year survival and these genes in the risk model. Among them, *CANX*, *BAG3*, *DIRAS3*, *PARP1*, and *PRKCD* are more consistent in both TCGA-LAML and GSE37642 cohorts. This is partly a reflection of the lower efficiency of TCGA-LAML cohort when compared to GSEA cohorts. Additional data could help validate and optimize the model.

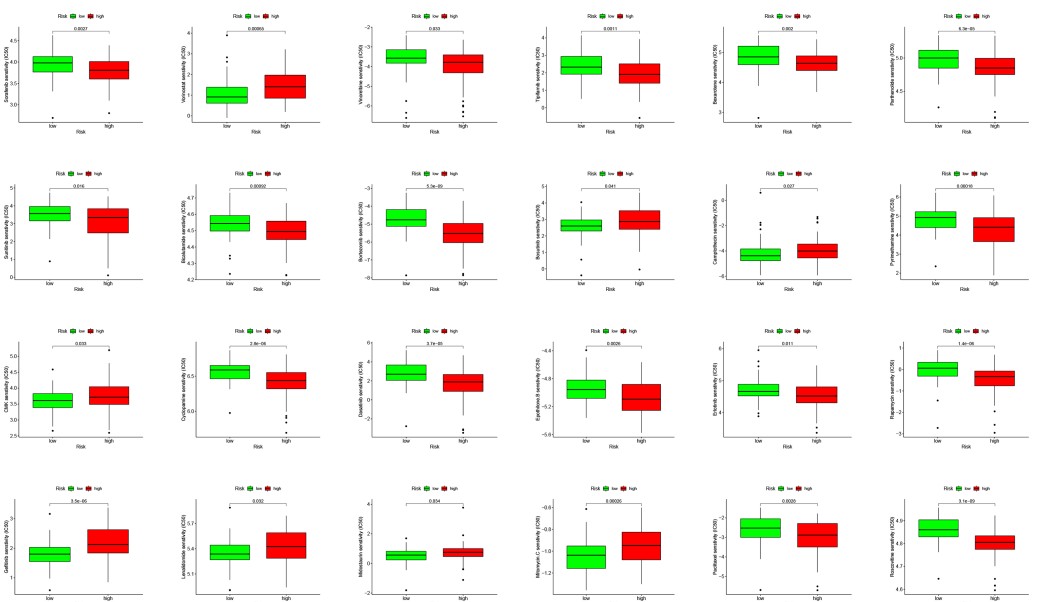

**Figure 7  Drug sensitivity analysis to drugs of high- and low-risk subgroups.**

In addition, we analyzed the relationship between autophagy genes and immune infiltrating cells in the model, and the results showed that the high-risk subgroup had a higher level of StromalScore, ImmuneScore, and certain immune cell types compared to the low-risk subgroup, indicating that the model might have a special immune signature. Moreover, the expression level of immune checkpoint genes (*CTLA4* and *CD274*) in patients with higher risk was higher than low-risk subgroups, suggesting this model provides more information for immune therapies like stratifying patients who are more sensitive for *CTLA4* and *CD274* immune therapies. Consequently, we xplored the relationship between AML and tumor environment in the TCGA-LAML cohort. We found StromalScore could not predict prognosis but higher ImmuneScore had a slightly better survival while age is a significant factor that influencing Stromal Score and Immune Score in TCGA-LAML cohort. However, for mast cells resting and NK cells activating, subgroups with relatively high- or low level had a significant different survival. Those findings supported that AML patients might respond to immune therapies and our model might help their clinical applications. On the other hand, the pathway enrichment in high-risk subgroup in GSEA showed the top five enriched pathways –KEGG CHEMOKINE SIGNALING PATHWAY, KEGG CELL ADHESION MOLECULES CAMS, KEGG CYTOKINE CYTOKINE RECEPTOR INTERACTION, KEGG HEMATOPOIETIC CELL LINEAGE, and KEGG INTESTINAL IMMUNE NETWORK FOR IGA PROC. This together with the immune environment relationship, these results help clarify the interactions among autophagy and other signaling pathways in AML.

*DIRAS3,* one important gene in our risk score model, is an imprinted tumor suppressor gene that also plays a very vital role in ovarian and breast cancer (*Sutton et al., 2019a*; *Peng et al., 2018*; *Sutton et al., 2019b*). *PRKCD* is a pro-apoptotic kinase, and some

miRNAs can regulate tumors by targeting *PRKCD* (*Zhang, Xu & Dong, 2017*; *Yao et al., 2015*; *Ke et al., 2013*). *VAMP3* is a member of the vesicle-associated membrane protein (VAMP)/synaptobrevin family (*Sneeggen et al., 2019*; *Chen et al., 2019*; *Pontes et al., 2006*; *Caronni et al., 2018*). Consistent with these studies, our research shows that these genes are potential therapeutic targets for postoperative diseases caused by microglial activation.

However, this study has some limitations. First, our study is mainly based on TCGA data, and most of the patients are white or Asian and we should be cautious to extend our findings to patients of other races. Second, our study is a retrospective analysis, and prospective studies are necessary to verify the results. Third, the AML datasets do not have complete clinical information, which may reduce the statistical validity and reliability. Finally, verification of our model *in vitro* or *in vivo* would be beneficial.

Overall, we constructed a prognostic model of 10 autophagy-related genes through the TCGA database and verified them through the GEO database. Our results complement the existing prognostic models and can be used as potential biomarkers for AML. In addition, we provide new views on the role of autophagy genes in AML, and these autophagy genes may also be applied in clinical adjuvant therapy.

**Abbreviations**

| | |
|---|---|
| **AML** | Acute myeloid leukemia |
| **KEGG** | Kyoto Encyclopedia of Genes and Genomes |
| **GO** | gene ontology |
| **DEGs** | differentially expressed genes |
| **FC** | fold change |
| **GSEA** | gene set enrichment analysis |
| **HR** | hazard ratio |
| **ROC** | receiver operating characteristic |
| **AUC** | area under the ROC curve |
| **LAML** | Acute myeloid leukemia |
| **LASSO** | least absolute shrinkage and selection operator |
| **TCGA** | The Cancer Genome Atlas |
| **GEO** | Gene Expression Omnibus |

### Funding

This work was supported by the Medical and health research projects in Hainan Province (Grant No. 2001320243A2009). The funders had no role in study design, data collection and analysis, decision to publish, or preparation of the manuscript.

### Grant Disclosures

The following grant information was disclosed by the authors:
the Medical and health research projects in Hainan Province: 2001320243A2009.

### Competing Interests

The authors declare there are no competing interests.

## Author Contributions

- Li Huang conceived and designed the experiments, performed the experiments, analyzed the data, prepared figures and/or tables, and approved the final draft.
- Lier Lin conceived and designed the experiments, performed the experiments, analyzed the data, prepared figures and/or tables, authored or reviewed drafts of the paper, and approved the final draft.
- Xiangjun Fu conceived and designed the experiments, performed the experiments, prepared figures and/or tables, and approved the final draft.
- Can Meng conceived and designed the experiments, performed the experiments, prepared figures and/or tables, authored or reviewed drafts of the paper, and approved the final draft.

## Data Availability

The R script is available in the Supplemental Files.

## Supplemental Information

Supplemental information for this article can be found online at http://dx.doi.org/10.7717/peerj.11968#supplemental-information.

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
