# Peer review of "Development and validation of a novel survival model for acute myeloid leukemia based on autophagy-related genes"

_PeerJ, doi:10.7717/peerj.11968_

## Round 0.1 · original submission · Major Revisions

Your manuscript "Development and validation of a novel survival model for acute myeloid leukemia based on autophagy-related genes" has been assessed by our reviewers. Although it is of interest, we are unable to consider it for publication in its current form. The reviewers have raised a number of points that we believe would improve the manuscript and may allow a revised version to be published in PeerJ.

Reviewer 1 ·

Basic reporting

no comment

Experimental design

no comment

Validity of the findings

no comment

Additional comments

It's an interesting paper based on bioinformatics analysis. I have several suggestions to improve the paper.
1. What's the c-index of the signature? Please provide it in the revision.
2. The signature performed well in the training set, however, in the validation it didn't work as well. It's better to randomly divide the training set as inner training and inner validation set, or use the 10-fold cross-validation in the training set to construct the signature, which may achieve a better c-index.

Reviewer 2 ·

Basic reporting

The English writing needs to be polished for the wording is inaccurate. For example, in the abstract ,"The results showed that the high-risk subgroup was prone to the poor prognosis of the training TCGA-LAML cohort and the validation GSE37642 cohort." the word prone is misused.
Picture pixels are very low.

Experimental design

The author needed to clarify the details how they merged two datasets GSE37642-GPL96 and GSE37642 GPL570 for they were from the different platform.

Validity of the findings

The prediction accuracy is low in validation dataset and more datasets are needed to validate this signature.

Additional comments

1、There is great difference in prediction accuracy in prediction and validation dataset for the auc is above 0.8 in construction dataset but the auc is above 0.5 in construction dataset. This is due to overftting effect in the construction dataset. More datasets are needed to validate this signature.

2.The author chose cutoff value using the median value and please use another way to choose cutoff value.

3. Nomogram is often included clinicopathologic parameters. And it seems unnecessary since there are very few clinicopathologic parameters in both datasets. What is the prediction accuracy of the nomogram compared with the signature?


4.The author needed to clarify the details how they merged two datasets GSE37642-GPL96 and GSE37642 GPL570 for they were from the different platform.

Reviewer 3 ·

Basic reporting

I have listed a few minor format issues that need to be addressed or clarified.
1. Please add LAML, LASSO, TCGA to the abbreviation.
2. In Line 128, please specify if you use a two-sided test or a one-sided test.
3. In Line 97, it misses a multiplication sign between the expression level and the beta coefficients. Please specify the right form of the expression level (linear scale or log-transformed scale?).
4. What causes NRG2 to have such a high HR compared to others in Fig 1A?
5. The label of the x-axis is missing in Fig. 1E.
6. Please specify the variance explained by both two components in Fig 6A.
7. Please explain why a cutoff of 65 years old is used in Fig. 6D?
8. Please make sure that Fig. 6B and 6F are consistent in the confidence interval band and the location of p values.

Experimental design

Most of the comments are related to the statistical analyses performed to train the 10-gene signature for reproducibility proposes.
1. To facilitate reproducibility, it is highly recommended to share your R code used in the analysis.
2. In Line 71, how did you identify these 35 autophagy genes? Is it from a database, a published article, or software?
3. In Line 86, please provide the selection criteria in texts or a diagram on how to retrieve 173 samples. Currently, the TCGA website shows 200 samples available.
4. Please cite all the R packages used in Section 2 Materials and Methods including glmnet, survminer and etc.
5. The sample size in Fig. 2A is 140 while it says 173 in Line 86. Please explain the inconsistency.
6. In Section 3.3 Clinical correlation analysis, why are only age and gender fitted in the multivariate analysis? How about other available clinical variables in the TCGA-LAML cohort? The same question is applied to the GSE3642 cohort.
7. In Line 123, please specify the number of folds used in the cross-validation, and the criteria used to identify the optimum parameter lambda.

Validity of the findings

1. Please include a table to compare the detailed clinical baseline characteristics between the training set and the validation set (TCGA and GEO datasets).
2. Is it possible that we can compare this 10-gene risk model with other published models?

Reviewer 4 ·

Basic reporting

There are grammatical errors and typos in this manuscript for example:
- Specifically, high risk subgroup were likely to have a worse survival.
- Last, the distinct drug sensitivity between patients with high- or low-risk provide further information for precise treatment for different groups .
- These findings can provide new insights for the ...

More literature review should be added in terms of related works on bioinformatics analysis.

Quallity of figures should be improved.

Experimental design

- The authors merged 2 different datasets without concern about batch effect removal.

- ROC curve has been used in previous bioinformatics studies such as PMID: 33260643, PMID: 31987913, and PMID: 33539511. Therefore, the authors are suggested to refer to more works in this description to attract a broader readership.

- Why did the authors need to run to 1000 fold CV?

- Source codes should be provided for replicating the methods.

Validity of the findings

- The authors mentioned "However, this study has some limitations", but they only showed a limitation as "Our analysis results have not been verified by in vitro or in vivo experiments". Thus, what are the other limitations?

Additional comments

No comment.

---

## Round 0.2 · Minor Revisions

The reviews are in general favorable and suggest that, subject to minor revisions, before your manuscript could be suitable for publication. Please consider these suggestions, and I look forward to receiving your revision.

Reviewer 3 ·

Basic reporting

For some reason, all the newly added figures and tables are uploaded but not the revised figures mentioned in the rebuttal letter. Could you make sure those are uploaded to the revision submission?

1. The label of the x-axis is still missing in Fig. 1E.
2. Please specify the variance explained by both two components in Fig 6A. Please note this comment is not about the legend. But the variance explained by the PCA component.
3. Please make sure that Fig. 6B and 6F are consistent in the confidence interval band and the location of p values. Again, I don't see any confidence interval bands are added.

Experimental design

Thank you for uploading the R code.

1. The number of folds is 10 not 1000 given the code provided.
fit <- glmnet(x, y, family = "cox", maxit = 1000)
maxit is the maximum number of iteration not the number of folds.

Validity of the findings

no comment

Additional comments

Thank you for addressing all my comments.

Reviewer 4 ·

Basic reporting

No comment.

Experimental design

No comment.

Validity of the findings

No comment.

Additional comments

My previous comments have been addressed well.

---

## Round 0.3 · accepted · Accept

I am pleased to inform you that your manuscript "Development and validation of a novel survival model for acute myeloid leukemia based on autophagy-related genes" has been approved for production and accepted for publication in PeerJ.